# Proteomic Analysis of Dupuytren’s Contracture-Derived Sweat Glands Revealed the Synthesis of Connective Tissue Growth Factor and Initiation of Epithelial-Mesenchymal Transition as Major Pathogenetic Events

**DOI:** 10.3390/ijms24021081

**Published:** 2023-01-05

**Authors:** Claudia Griselda Cárdenas-León, Kristina Mäemets-Allas, Mariliis Klaas, Katre Maasalu, Viljar Jaks

**Affiliations:** 1Department of Cell Biology, Institute of Molecular and Cell Biology, University of Tartu, 51010 Tartu, Estonia; 2Department of Traumatology and Orthopedics, Institute of Clinical Medicine, University of Tartu, 51010 Tartu, Estonia; 3Clinic of Traumatology and Orthopedics, Tartu University Hospital, 51010 Tartu, Estonia; 4Dermatology Clinic, Tartu University Clinics, 50411 Tartu, Estonia

**Keywords:** Dupuytren’s contracture, sweat gland, CTGF, E-cadherin, epithelial-mesenchymal transition, extracellular matrix

## Abstract

Dupuytren’s contracture (DC) is a chronic and progressive fibroproliferative disorder restricted to the palmar fascia of the hands. Previously, we discovered the presence of high levels of connective tissue growth factor in sweat glands in the vicinity of DC nodules and hypothesized that sweat glands have an important role in the formation of DC lesions. Here, we shed light on the role of sweat glands in the DC pathogenesis by proteomic analysis and immunofluorescence microscopy. We demonstrated that a fraction of sweat gland epithelium underwent epithelial-mesenchymal transition illustrated by negative regulation of E-cadherin. We hypothesized that the increase in connective tissue growth factor expression in DC sweat glands has both autocrine and paracrine effects in sustaining the DC formation and inducing pathological changes in DC-associated sweat glands.

## 1. Introduction

Dupuytren’s contracture (DC) is a chronic and progressive fibroproliferative disorder restricted to the palmar fascia of the hands. A characteristic symptom of the late phase of the disease is the irreversible flexion contracture of one or more fingers with significant loss of their function [1]. The prevalence rates of this disease depend on the population group analyzed and are significantly higher in middle-aged or older men of Northern European descent [2]. In addition, metabolic factors such as diabetes mellitus and excessive alcohol consumption constitute the main risk factors for development of DC [3]. Other risk factors include smoking, increasing age, male gender, epilepsy, and genetic predisposition [4,5].

High-throughput analyses of DC tissue have revealed valuable insight into the pathogenesis of this disease. Recent results from single-cell RNA sequencing outlined a population of podoplanin (PDPN)+ pathogenic fibroblasts, which were thought to drive the DC pathogenesis via several mechanisms that involved modulation of inflammation, extracellular matrix (ECM) deposition and cell proliferation [1]. Additionally, studying epigenetic and transcriptional profiles of primary human myofibroblasts obtained from DC nodules pointed to the acetyltransferase CREBBP/EP300 as an important regulator of contractility and ECM production through the control of H3K27 acetylation in the profibrotic genes, actin alpha 2, smooth muscle (ACTA2) and collagen type I (COL1A1) [6]. 

In our previous work, we discovered the presence of high levels of connective tissue growth factor (CTGF) in sweat glands located in the vicinity of DC nodules. As CTGF has been implicated in the pathogenesis of DC we proposed a model where sweat glands and other structures such as small blood vessels participate in the DC pathogenesis by contributing factors that sustain the proliferation of myofibroblasts that eventually forming the DC nodule [7]. To expand our knowledge about the potential role of sweat glands in DC we analyzed the proteomic profile of DC-associated sweat glands to elucidate their potential molecular mechanisms by which these could participate in the pathogenesis of DC. We found more than 600 proteins that were differentially expressed in DC sweat glands. A significant downregulation of E-cadherin (CDH1), α-catenin (CTNNA1), β-catenin (CTNNB1), and p120 (CTNND1) was detected, as well as upregulation of mimecan (OGN), decorin (DCN), thrombospondin-4 (THBS4) and connective tissue growth factor (CTGF), among other proteins. We used immunofluorescence microscopy to corroborate our proteomic analysis and demonstrated that epithelial-mesenchymal transition (EMT) occurring in sweat glands connected to increased synthesis of CTGF potentially participating in the progression of DC.

## 2. Results

To collect material for proteomics analysis, we followed the protocol previously established by Gao et al. (2014) [8] to isolate sweat glands from human palmar skin. The Appendix A shows a representative micrograph of an isolated sweat gland. The proteome analysis identified 4041 proteins, of which 641 were immunoglobulins and 128 were cytokeratins, that were eliminated from further analysis. After data processing, we identified 619 differentially expressed proteins, of which 364 were downregulated and 255 upregulated. The list of significantly regulated proteins is presented in Appendix A. Functional analyses revealed that the top two enriched biological process were “ECM organization”, and “supramolecular fiber organization suggesting the presence of fibrosis in the sweat glands (Figure 1A). While the top enriched signaling pathways were “protein processing in endoplasmic reticulum”, “ECM-receptor interaction”, “focal adhesion” and “PI3K-Akt” signaling pathways (Figure 1B). A detailed pathway analysis outlined a connected network consisting of “adherens junction”, “ECM-receptor interaction” and “PI3K-AKT” signaling pathways (Appendix A).

We noted the upregulation of several non-structural ECM proteins: mimecan (increased 129×), periostin (POSTN, 81×), thrombospondin-4 (21×), tenascin-C (TNC, 19×), cartilage oligomeric matrix protein (COMP, 16×), and secreted protein and acid rich in cysteine (SPARC, 7×). Increased levels of proteoglycans such as dermatopontin (DPT, 31×), asporin (ASPN, 23×), lumican (LUM, 11×), decorin (9×) and fibronectin (FN1, 20×) were detected. Similarly, the levels of several structural ECM proteins such as collagen type I (46×), collagen type II (COL2A1, 46×), collagen triple helix repeat containing 1 (CTHRC1, 22×), collagen type XII (COL12A1, 16×), collagen type XVIII (COL18A1, 15×) and collagen type IV (COL4A1, 8×) were upregulated. Interestingly, we noted the upregulation of transforming growth factor beta induced factor (TGFBI or BGH3), and connective tissue growth factor by 3.65 and 2.46-fold, respectively. At the same time, we detected the downregulation of epithelial markers E-cadherin (decreased 0.04×), α-catenin (0.63×), β-catenin (0.28×) and p120 (0.49×) (Figure 1C,D). In addition, we detected the upregulation of integrin subunit alpha V (ITGAV, 3.29×), Ras homolog gene family, member C (RhoC, 1.78×), and Ras-related proteins (Rab23, 5.41×; Rab8A, 1.47×), while Ras suppressor-1 was downregulated (RSU1, 0.46×). 

The increase of CTGF in DC samples was detected at the mRNA level also in an earlier work [7]. To confirm our findings, we first evaluated the expression of CTGF by immunofluorescence staining. To visualize the sweat glands, we co-stained the samples with keratin 15 (K15) and the basal membrane marker Collagen IV (COL41A) antibodies. CGTF protein was detected in both control and DC sweat glands in acini but not sweat glands ducts (Figure 2A–D, Appendix A). However, the CTGF level was 2.78 ± 0.53-fold higher in the acini of DC sweat glands, well corroborating the results of proteomic analysis (Figure 2I). Next, we evaluated the presence of decorin in the K15-positive acini of sweat glands. As expected, the decorin protein was prominently present in DC nodule and surrounding areas of sweat glands, while control samples showed only a weak decorin signal (Figure 2E–H, Appendix A). Furthermore, the decorin protein level was 2.66 ± 0.14-fold higher in DC samples when compared to control sections (Figure 2J).

To study the localization of β-catenin and E-cadherin, epithelial markers that act as structural proteins as well as important intracellular signaling molecules, we co-stained control and DC samples with antibodies recognizing CTGF, β-catenin and E-cadherin (Figure 3). We observed that β-catenin was present in sweat gland ducts and acini in the control samples as well as in the DC samples, and E-cadherin co-localized with β-catenin in control samples regardless of the signal strength (Figure 3E,F,I,J,M,N). Whereas in the DC samples, the level of E-cadherin was markedly decreased and the colocalization of both proteins was observed only in acini near sweat gland lumens (Figure 3G,K,O), and ducts (Figure 3H,L,P). Furthermore, both β-catenin and E-cadherin were virtually absent in basal areas of DC sweat glands acini, but not in ducts. Consistent with the proteomic analysis, the fluorescent intensity analysis revealed a significant E-cadherin downregulation (0.31 ± 0.13×) in DC sections compared to control samples (Figure 3Q). There was also a notable tendency towards reduction in the β-catenin levels (0.6×) however, this change was not statistically significant (Figure 3R). The E-cadherin/β-catenin ratio was also reduced in DC samples (0.15) when compared to control samples (0.44) (Figure 3S).

Loss of E-cadherin has been linked to EMT where cells with an epithelial phenotype progressively lose epithelial markers and gain mesenchymal markers, acquiring a more mesenchymal phenotype [9]. We hypothesized that cells in DC sweat glands gradually pass through EMT and convert to a mesenchymal-like phenotype. Therefore, we stained control and DC samples with antibodies recognizing the potential markers for EMT—the smooth muscle actin (SMA) [10] and vimentin (VIM) [11] and co-stained the samples with an antibody recognizing β-catenin to identify epithelial cells in sweat glands (Figure 4). We found that in normal sweat glands, proper compartmentalization of mesenchymal and epithelial markers was retained: vimentin was localized to the connective tissue surrounding the acini, and SMA localized to the smooth muscle layer encircling the β-catenin -positive acinar epithelial cells (Figure 4A,C,E,G). In a subset of DC-associated sweat glands, such a clear separation of epithelial and mesenchymal markers was lost, and one could observe the colocalization of the epithelial marker β-catenin and mesenchymal markers SMA and vimentin (Figure 4B,D,F,H, Appendix A), which suggests the occurrence of EMT in the acini of the DC-associated sweat glands.

## 3. Discussion

Myofibroblasts are the main cellular component of DC nodules, and the main source of production and deposition of collagen fibers in ECM [11]. So far, most of the studies addressing the pathogenesis of DC have focused on fibroblasts and myofibroblasts that directly participate in the formation of pathological nodules [11,12]. Nevertheless, we have shown previously that additional structures like blood vessels and sweat glands may have a prominent role in the development of DC lesions by providing pro-proliferative and profibrotic signals [7]. On the other hand, sweat glands have attracted the attention of researchers since they can contribute to skin wound healing and regeneration by activation of a specific stem cell compartment [13,14,15]. 

To shed further light on the role of sweat glands in the DC pathogenesis we performed a proteomic analysis of sweat glands isolated from DC tissue. Our analysis confirmed our previous result that CTGF is the main growth factor that was expressed at higher levels in DC-associate sweat glands. CTGF can act both in an autocrine and a paracrine manner. While TGF-β is a powerful activator of fibrosis and EMT, CTGF can initiate fibrosis independently of TGF-β signals in a paracrine manner by activating the proliferation and differentiation of myofibroblasts [16]. In addition, during autocrine signaling, CTGF can induce the expression of TGF-β at the transcriptional level [17]. Additionally, vice versa, TGF-β treatment upregulated CTGF expression in human primary granulosa cells, as well as nucleus pulposus of the intervertebral disc cells, via SMAD and ERK1/2, and SMAD3/activator protein (AP-1) signaling pathways, respectively [18,19]. Although, depending on the cell type, several signaling pathways, including p38 MAPK, JNK, STAT3 and PKC, may be involved in the upregulation of CTGF expression induced by TGF-β [17,20,21]. In line with the anticipated activation of TGF-β signaling, our proteomic analysis detected an upregulation of TGFBI in DC-associated sweat glands. 

Our proteomic analysis showed downregulation of epithelial markers E-cadherin, α-catenin, β-catenin, and p120. Due to the loss of E-cadherin, some of β-catenin could be freed that may exert its role as a transcription factor and activate Wnt target genes and different downstream signaling pathways such as PI3K/AKT and RhoA, where translocation of β-catenin to the nucleus of is not always required [22]. Although we were unable to detect nuclear localization of β-catenin, downregulation of E-cadherin suggests initiation of EMT [23,24]. In addition, we observed the downregulation of Ras suppressor-1, and the upregulation of Ras superfamily members Rab23, Rab8A, and RhoC as well as several integrins, including ITGAV in our proteomic results, which, taken together suggest the activation of a non-canonical TGF-β signaling pathway, most probably through the Ras signaling pathway [25,26,27]. In addition, the association between sweat gland fibrosis and EMT has previously been described in morphea [28], a type of localized scleroderma and skin disease involving fibrosis in the dermis and adipose tissue [29]. Similar to our findings in DC, the expression of TGF-β, SMA, and fibronectin was increased in the dermis of the morphea samples, while the expression of E-cadherin was decreased and the expression of Snail1—a prominent mediator of EMT—was increased in the dermal eccrine glands [28]. EMT and fibrosis have been linked in the skin fibrotic pathologies, such as keloids, as well as in fibrotic pathologies taking place in other tissues [30,31]. 

The occurrence of EMT in DC-associated sweat glands was further confirmed by the colocalization of the mesenchymal and epithelial markers marker vimentin and β-catenin in the sweat gland epithelial compartment. Interestingly, no increase in vimentin or SMA were detected by our proteomic analysis. However, this notion raises the hypothesis that sweat gland cells which acquired a partially mesenchymal phenotype could release signals that may stimulate tissue fibrosis and even initiate fibrosis inside the sweat glands themselves turning these into a source of new myofibroblasts. Consistent with earlier findings, microvessels have been suggested to be an external source of myofibroblast progenitors in DC [32]. 

The most recognized ECM proteins identified in DC tissues are collagens [33]. According to our proteomic results, we identified the upregulation of various collagens type I¬-VI, VIII, XII, XIV, XVIII, XXVIII and collagen triple helix repeat containing 1 type. Nevertheless, also several non-structural ECM proteins such as mimecan, periostin, decorin and TGFBI were upregulated in DC associated sweat glands. Mimecan, periostin, decorin and TGFBI are proteins that have been shown to interact with fibrillar collagens, growth factors and growth factor receptors and thus influence ECM assembly, cellular growth, and migration [1,2,3]. TGFBI has been shown to be induced by TGF-β [34] and can affect heart and renal fibrosis [35,36]. Moreover, decorin modulates TGF-β activity, and at the same time, TGF-β could regulate periostin expression, the latter of which has been shown to act as a central element in EMT induced by TGF-β [37]. Furthermore, the decorin–TGF-β interaction modulates the activity of the latter, however the biological consequences of this interaction are not yet fully understood [38]. The upregulation of tissue inhibitor of metalloproteinases 2, TGF-β, and decorin in DC samples has been shown before, arguing that these three proteins could play an essential role in the assembly of pathological ECM [38]. Thus, we corroborate these earlier findings and propose that sweat glands could be a source of decorin in DC formation. 

In conclusion, we envision that CTGF in DC-associated sweat glands acts in an autocrine manner inducing TGF-β. This establishes a positive feedback loop that drives the endogenous CTGF expression and contributes to EMT in acinar cells illustrated by downregulation of E-cadherin [16]. CTGF acts also in paracrine manner and facilitates the differentiation of myofibroblasts in the DC nodule as well as may stimulate the synthesis of TGF-β in the DC nodule closing thereby the positive feedback loop. Indeed, local production of TGF-β has been shown to take place in myofibroblasts and fibroblasts during all stages of DC [39]. In parallel, several ECM molecules secreted by sweat glands have been shown to have a role in DC or in fibrosis and may thus further facilitate formation of a DC nodule. The proposed role of sweat glands in DC is summarized in Figure 5.

The main limitations of our work include the heterogenous degree of fibrosis in the collected DC samples, the relatively low number of samples, and the variability between patients. Additionally, the proteomic results may be obfuscated by the presence of several cell types in the protein preparation as there exist different cell types in the sweat gland compartment: the secretory coil or acini and the duct, are formed by different cells, myoepithelial, basal clear, and apical dark cells in acini, as well as epithelial cells in the duct [15,40]. Due to these limitations, further studies are needed to precisely identify the role of sweat glands in DC formation.

## 4. Materials and Methods

### 4.1. Collection of Tissue Sample and Sweat Glands

Nodular tissue samples from Dupuytren’s contracture (DC samples) were obtained from biopsies taken from twelve patients with DC with extensive, stage 2 or 4 fibrosis of the palmar fascia undergoing open radical palmar fasciectomy. Normal palmar fascia (control samples) were obtained from nine patients unaffected by DC undergoing open carpal tunnel release surgery. All samples were collected at the Tartu University Hospital, Surgery Clinic between January to June 2022 (Permit 335/T-1 from the Committee for Human Studies, University of Tartu), and the procedures were performed in accordance with The Declaration of Helsinki and written informed consent was obtained from the patients. In total twenty-seven samples were collected: twelve control samples and fifteen DC samples, of which, nine samples per condition were used to collect sweat glands for the proteomics experiment, while three control samples and six DC samples were used for immunofluorescence analysis. 

The nodular tissue was separated from the DC chords, embedded in O.C.T compound (Sakura Finetek) and stored at −80 °C until further analysis. Six DC samples and three control samples were used for immunofluorescence analysis. Eighteen samples from eighteen different patients were used to collect sweat glands. For sweat gland separation, subcutaneous fat and DC chords were removed under aseptic conditions, the remaining tissues were minced into small pieces and digested overnight with 0.1% Collagenase I (Thermo Fisher Scientific, Grand Island, NY, USA) at 37 °C in an incubator with 5% CO_2_. On the next day, sweat glands were isolated with a pipettor under an inverse microscope [8]. Sweat glands were collected into separate tubes (per patient) and stored at −80 °C until further analysis. Appendix A shows data on donor age, sex, and experiment in which the samples were used.

### 4.2. Proteomic Analysis

Sweat glands from DC tissue samples (*n* = 9) and sweat glands isolated from normal palmar skin (*n* = 9) were denatured, alkylated, and digested as previously described [41]. Samples were injected into Thermo ScientificTM DionexTM UltimateTM 3000 RSLCnano systems equipped with NCS-3500RS (microflow) module and Q Exactive Plus—OrbitrapTM spectrometer (Thermo Fisher Scientific) operated in Data-Independent Acquisition (DIA) mode for peptide and protein quantification. Raw data was processed and analyzed by DIA-NN software [42] using default parameters. Q value (FDR) cut-off on precursor and protein level was applied 1%. All selected precursors that passed the filters were used for quantification. Proteomic data was analyzed using the Perseus software platform version 1.6.15.0 (www.perseus-framework.org (accessed on 10 November 2022) [43]). Samples with few data (names in manuscript: Control 7–9, DC 7–9; names in protein dataset: N1, N2 and N8, DD0, DD5 and DD7) as well as keratins, and immunoglobulins were removed. Data was filtered for at least 70% of the total valid values, and missing values were imputed from normal distribution width 0.3, down-shift 1.8. Up- or down-regulation was considered significant when *p* value < 0.05. The Kyoto Encyclopedia of Genes and Genomes (KEGG) mapper was used to visualize the signaling pathways involved, and enrichment analysis was performed using EnrichR tool (www.maayanlab.cloud/Enrichr/ (accessed on 10 November 2022)).

### 4.3. Immunofluorescence Analysis

For immunofluorescence analysis, 10 μm thick tissue cryosections were fixed in 4% paraformaldehyde solution for 10 min, rinsed with PBS, and permeabilized with 0.2% Triton-X-100 for 15 min. Samples were then blocked with 4% normal donkey serum for 1h at RT, incubated with primary antibodies overnight at 4 °C, and incubated with secondary antibodies for 1h at RT. The antibodies and their used dilutions are listed in Appendix A. Nuclei were counterstained with DAPI (0.1 μg/mL). Images were acquired with the Olympus BX-71 fluorescence microscope and processed using Hokawo v2.1 (Hamamatsu Photonics) software. The integrated density of fluorescent signals for selected antibodies was analyzed using ImageJ software.

### 4.4. Statistics

The immunofluorescence experiments as well as the intensity of fluorescent signals were performed in triplicate. The results are presented as the mean ± standard deviation. Student’s *t*-test was used to analyze the data. The differences were considered statistically significant when *p* < 0.05.

## Figures and Tables

**Figure 1 ijms-24-01081-f001:**
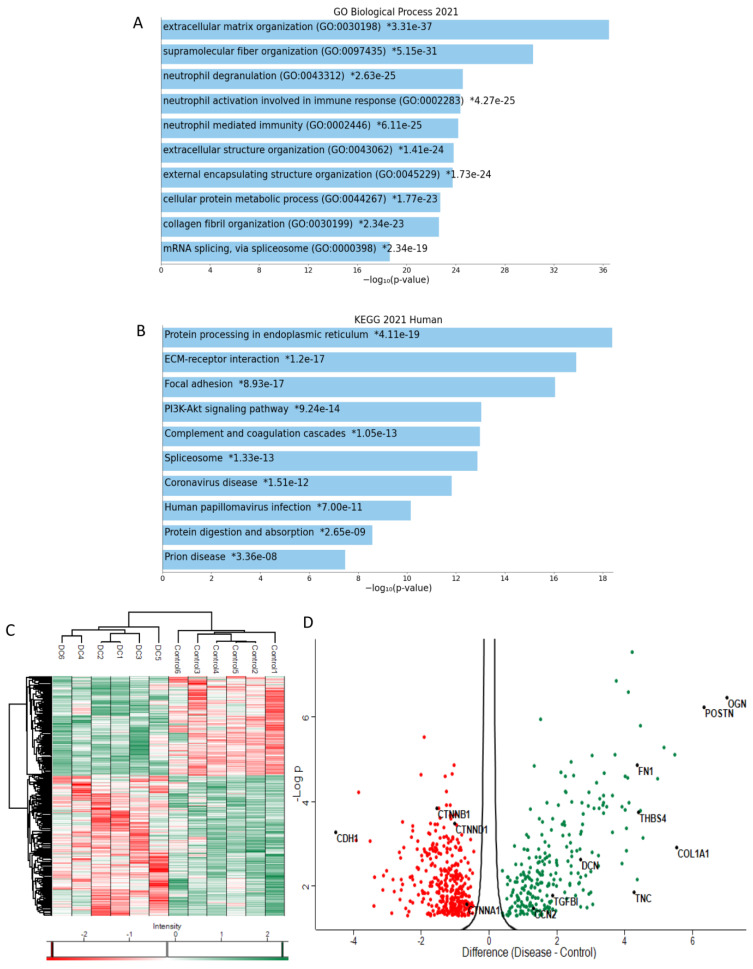
Proteomic analysis of sweat glands obtained from Dupuytren’s contracture. Enrichment analysis GO Biological process (**A**) and Kyoto Encyclopedia of Genes and Genomes pathways related (**B**). The bar chart shows the top 10 enriched terms, along with their corresponding *p*-values (* *p* < 0.05). Heat map (**C**) and volcano plot (**D**) of proteomic analysis.

**Figure 2 ijms-24-01081-f002:**
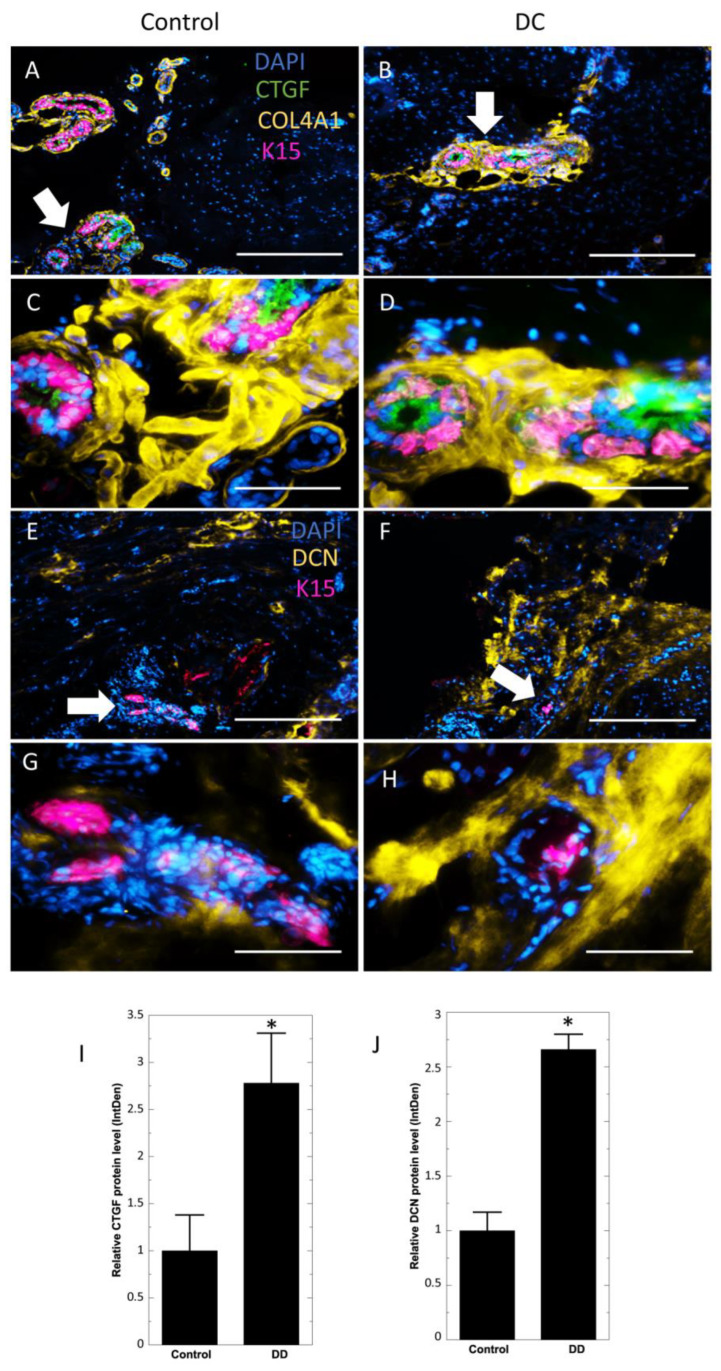
Sweat glands of Dupuytren’s contracture samples present high levels of CTGF and decorin. Immunofluorescence analysis of control and DC samples were stained for keratin 15, CTGF, and Collagen IV (**A**–**D**), and for keratin 15 and decorin (**E**–**H**). Scale bar: A, B, E, F—200 µm; C, D, G, H—50 µm. Arrows in A, B, E and F show the area amplified in C, D, G and H, respectively. Quantitative analysis of CTGF (**I**) and decorin (**J**) levels in control and DC samples. Results are presented as mean ± SD and compared to control samples. * *p* < 0.05.

**Figure 3 ijms-24-01081-f003:**
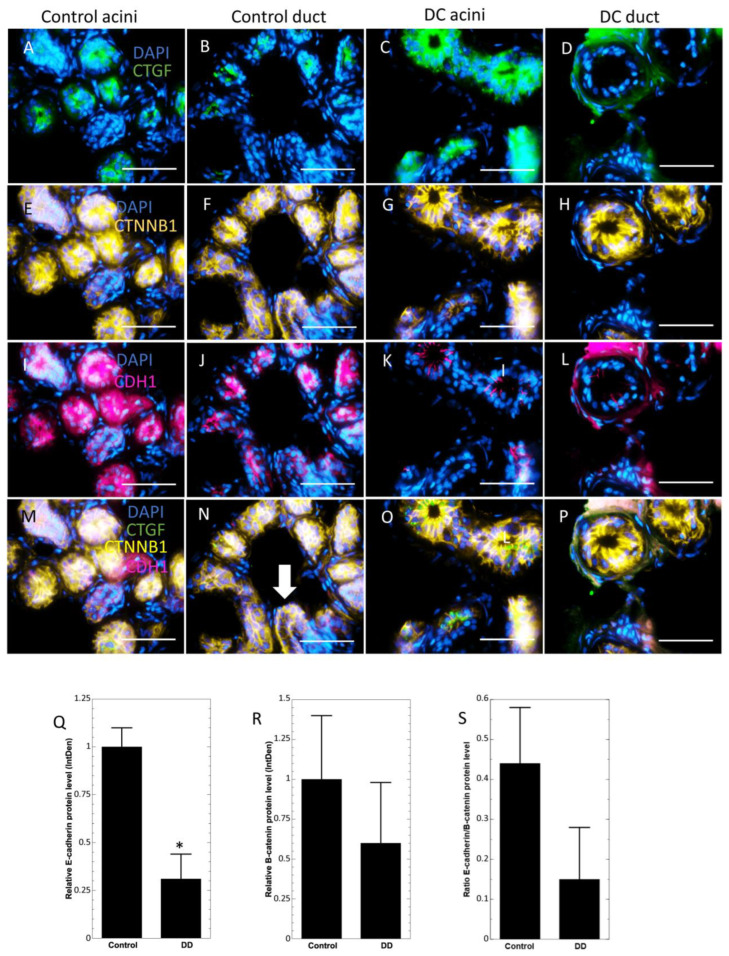
Sweat glands of Dupuytren’s contracture samples present low levels of E-cadherin. Immunofluorescence analysis of control sweat gland acini (**A**,**E**,**I**,**M**) and duct (**B**,**F**,**J**,**N**) DC sweat gland acini (**C**,**G**,**K**,**O**) and duct (**D,H**,**L**,**P**) samples were stained for CTGF (**A**–**D**), β-catenin (**E**–**H**), E-cadherin (**I**–**L**) and merged (**M**–**P**). E-cadherin and β-catenin colocalization (orange color). Scale bar: 50 µm. Quantitative analysis of E-cadherin (**Q**), β-catenin (**R**) levels in control and DC samples, and E-cadherin/β-catenin ratio (**S**). The duct in N is marked with a white arrow. Results are presented as mean ± SD and compared to control samples. * *p* < 0.05.

**Figure 4 ijms-24-01081-f004:**
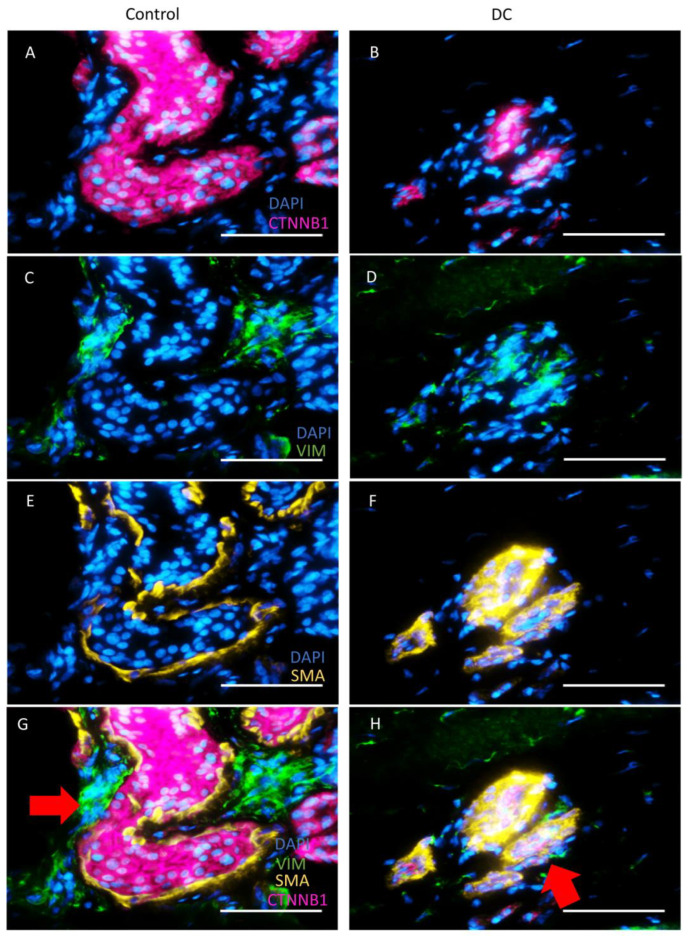
Dupuytren’s contracture samples with extensive stage of fibrosis present cells with intermediate epithelial-mesenchymal transition phenotype. Immunofluorescence analysis of control (**A**,**C**,**E**,**G**) and DC (**B**,**D**,**F**,**H**) samples were stained for β-catenin (**A**,**B**), vimentin (**C**,**D**), and smooth muscle actin (**E**,**F**) and merged (**G**,**H**). Red arrow in G indicates vimentin (green) localized to the connective tissue surrounding the acini, and SMA (yellow) localized to the smooth muscle layer encircling the β-catenin (red) -positive acinar epithelial cells. Red arrow in H indicates the colocalization (dark yellow) of the epithelial marker β-catenin (red) and mesenchymal markers SMA (yellow) and vimentin (green) in DC sweat glands. Scale bar: 50 µm.

**Figure 5 ijms-24-01081-f005:**
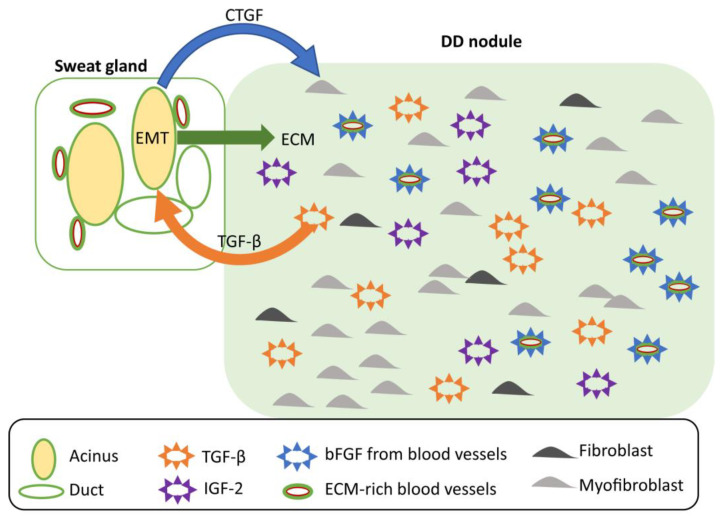
Schematic model summarizing the proposed role of sweat glands in the pathogenesis of DC. See discussion for more details.

## Data Availability

Proteomic data is freely available at https://repository.jpostdb.org/preview/352300491635e349b36227, access key 1810 (accessed on 31 October 2022).

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
