# Peer review of "Proteomic Analysis of Dupuytren’s Contracture-Derived Sweat Glands Revealed the Synthesis of Connective Tissue Growth Factor and Initiation of Epithelial-Mesenchymal Transition as Major Pathogenetic Events"

_ijms, 2023, doi:10.3390/ijms24021081_

Round 1
Reviewer 1 Report
The study descried in this manuscript aimed at investigating potential molecular and cellular mechanisms involved in the Dupuytren’s contracture disease (DD). Previously, the authors had reported that sweat glands associated with DC-palm tissue showed increased mRNA abundance of secreted protein connective tissue growth factor -encoding gene (CTGF) as compared to the control samples. In the current study they attempt to determine the functional implications of increased CTGF in DD pathology. Using the mass spectrometry-based proteomic analysis, they identified more than 600 proteins that were altered the sweat glands of patents as compared to those isolated from “control” subjects. Based on further bioinformatics-based pathway analysis and immunofluorescence studies, the authors suggest that CTGF may help to induce EMT in the sweat glands in a possible TGF-β-dependent manner, with potential implications for DD pathogenesis.
Comments
1. The authors should use full names of proteins/genes the first time introduced un the manuscript followed by their abbreviations in parenthesis.
2. The rationale of studying/looking at the sweat glands as a potential factor in DC pathogenesis should be clearly stated in the introduction. Additionally, a schematic should be considered to add showing the localization of these glands in the context of the control vs DC-derived tissue. Overall, this will help to better understand the study and described findings.
3. The term “expression” in the line 91, should be altered to a term like “protein abundance” so as not confuse that with that of mRNA transcript.
4. In figure 1, 1C and 1D have been switched in the figure legend.
5. The authors should change “CTNNB1” and “CDH1” , which denote genes’ names, to β-catenin and E-cadherin, if they are referring to their encoded products or proteins in this case.
6. Figure 3 shows sweat glands acini (control vs CD) and ducts (only DC shown, is there a reason for not including control?). The authors should indicate apical vs basal aspects of the acini structures if applicable, in Figures 2 and 3. In addition, some description of the phenotypes of the sweat glands between control and DC should be provided. At least based on Fig 3, it appears that whereas the control seems to show a filled lumen, the DC-associated ones show a clear hollow lumen. If so, are control undergoing EMT more so than the DC sweat glands? Although it is suggested that E-cadherin is reduced in the DC samples, the immunostaining indicates a seemingly proper localization, as opposed to the control, which seems to be mis-localized. The same is true for β-catenin staining. As the control-derived tissue is taken from patients with carpal tunnel syndrome, what are the evidence that these tissue/glands are not altered as compared to that in normal tissue?
7. In Figure 3, it will be great to add quantifications of the relative alteration in the co-localization of E-cadherin and β-catenin between control and DC conditions.
8. The data in Figure 4 that follow the protein abundance of SMA and Vimentin as EMT markers in control vs DC derived tissue would benefit from a better description what is shown and their implication. For example, relating to what is been pointed to/shown by the red arrows in panels G vs H. The data are somewhat confusing. Where similar analyses done with the CD-associated sweat glands and if so, did they show any differences? In this regard, the authors should discuss whether any alterations in the protein abundance and localization of these and other related gene-products were detected in the proteomics analysis. Positive or negative results, at least, should be alluded to in the discussion.
9. On lines 173 and 174, it is stated that there was an upregulation of TGFBI in DC-associated sweat gland. It was not clear whether this is the new data (proteomics data they have generated) or is it something previously published work they have stated there.
10. In the 3rd paragraph of the discussion (lines 175-180), where they have connected the β-Catenin pathway with PI3K/AKT, RhoA and E-cadherin, it will be good if they discuss these scenarios in the context of their proteomics pathway analysis data.
11. On lines 200 and 201, two “that’ need to be added after “TGFBI” and after “proteins”, an ‘N” should be added to “POST”.
12. An illustration of the signaling pathways uncovered in this study with all the feedback loops should be included.

Author Response
We wish to thank the reviewers for very constructive suggestions that helped to considerably increase the quality of the manuscript.
Please find the point-to-point answers to the reviewers’ criticisms below.
1. The authors should use full names of proteins/genes the first time introduced un the manuscript followed by their abbreviations in parenthesis.
Response: We agree and we changed this in the text.
2. The rationale of studying/looking at the sweat glands as a potential factor in DC pathogenesis should be clearly stated in the introduction. Additionally, a schematic should be considered to add showing the localization of these glands in the context of the control vs DC-derived tissue. Overall, this will help to better understand the study and described findings.
Response:
- In our previous work we noted the presence of sweat glands in the vicinity of the Dupuytren’s nodules. After closer study, we found that the sweat glands secreted increased amounts of CTGF – a factor well known to participate in the pathogenesis of this disease. During recent years we introduced multi-omics analysis in our lab and decided to study the DC-associated sweat glands in a more systemic manner to understand their potential role in the disease. We made an effort to substantiate the rationale of the study in a more comprehensive manner and added the following information in the introduction:
- “In our previous work, we discovered the presence of high levels of connective tissue growth factor (CTGF) in sweat glands located in the vicinity of DC nodules. As CTGF has been implicated in the pathogenesis of DC we proposed a model where sweat glands and other structures such as small blood vessels participate in the DC pathogenesis by contributing factors that sustain the proliferation of myofibroblasts that eventually form the DC nodule. To systematically expand our knowledge about the potential role of sweat glands in DC we analyzed the proteomic profile of DC-associated sweat glands to elucidate their potential molecular mechanisms by which these could participate in the pathogenesis of DC.”
- In addition, we added a new figure (Figure 5) where the location of sweat glands in DC is schematized, and the proposed role of sweat glands in DC is summarized. Furthermore, we added supplementary figures 7-10 showing normal tissue micrographs where the location of sweat glands is clear, and DC condition micrographs showing some markers that are present in sweat glands in the vicinity of fibrotic tissue.
3. The term “expression” in the line 91, should be altered to a term like “protein abundance” so as not confuse that with that of mRNA transcript.
Response: We agree, and we changed this in the text.
4. In figure 1, 1C and 1D have been switched in the figure legend.
Response: We changed this error in the text.
5. The authors should change “CTNNB1” and “CDH1”, which denote genes’ names, to β-catenin and E-cadherin, if they are referring to their encoded products or proteins in this case.
Response: We agree, and we changed this in the text.
6. Figure 3 shows sweat glands acini (control vs CD) and ducts (only DC shown, is there a reason for not including control?).
- The authors should indicate apical vs basal aspects of the acini structures if applicable,
- in Figures 2 and 3. In addition, some description of the phenotypes of the sweat glands between control and DC should be provided. At least based on Fig 3, it appears that whereas the control seems to show a filled lumen, the DC-associated ones show a clear hollow lumen. If so, are control undergoing EMT more so than the DC sweat glands?
- Although it is suggested that E-cadherin is reduced in the DC samples, the immunostaining indicates a seemingly proper localization, as opposed to the control, which seems to be mislocalized. The same is true for β-catenin staining. As the control-derived tissue is taken from patients with carpal tunnel syndrome, what are the evidence that these tissue/glands are not altered as compared to that in normal tissue?
Response:
- We apologize, the control ducts were omitted by mistake. We have added the duct micrographs to Figure 3 as panels 3 B, F, J, and N marked with a white arrow.
- As the basal aspect of the acinar cells is towards the outer contour and the apical aspects are located centrally, we decided not to include additional labels as this would have added unnecessary complexity to the figures.
- We thank the reviewer for pointing out the differences in the morphology of sweat glands. Indeed, the Lumen is more open in DS glands than in control glands in figure 3, however, not so in figure 2. In fact, browsing through the material we collected for both this and the previous article published in 2015, we could not find any definitive differences between the DC and control sweat glands. One can see also glands with open lumens in control sweat glands, perhaps at a lower frequency, though. As of now, we consider this as a positional effect, however, we agree that there may be present some degree of morphological bias. Since we are not sure that this is a clear phenomenon, we did not include any generalization in the text of the manuscript. “
- Indeed, it may seem that the localization is less organized in control ducts. However, this impression is occurring due to two simultaneous factors. First and foremost, the levels of both E-cadherin and β-catenin are markedly higher in control samples vs DC samples. Second, the cells are more tightly packed in control seat glands vs DC-associated glands. These two aspects create a more disorganized impression in control glands. In fact, at a closer look, both E-cadherin and β-catenin are well-organized at the proper locations near the plasma membrane also in the control samples and are present all throughout the gland in contrast to the DC-glands where the higher levels of both proteins are limited to more luminal areas and lateral cell-to-cell contacts. Indeed, both DC and carpal tunnel syndrome involve excessive fibrosis of the fibrous fascia immediately under the skin. Nevertheless, the transverse carpal ligament is located below the skin, the fibrosis there is a slow process with a low-level inflammation present and no active skin involvement has been observed. DC presents a more active process where the also lower dermal structures are often attached to the DC nodule so one would not expect any major alterations in the sweat glands isolated from carpal tunnel syndrome patients.
7. In Figure 3, it will be great to add quantifications of the relative alteration in the co-localization of E-cadherin and β-catenin between control and DC conditions.
Response: Yes, we agree to this. We experimented with a few image analysis algorithms, however, could not achieve any conclusive results, and thus the colocalization analysis is out of our reach at the moment.
8. The data in Figure 4 that follow the protein abundance of SMA and Vimentin as EMT markers in control vs DC derived tissue would benefit from a better description what is shown and their implication. For example, relating to what is been pointed to/shown by the red arrows in panels G vs H. The data are somewhat confusing. Where similar analyses done with the CD-associated sweat glands and if so, did they show any differences? In this regard, the authors should discuss whether any alterations in the protein abundance and localization of these and other related gene-products were detected in the proteomics analysis. Positive or negative results, at least, should be alluded to in the discussion.
Response: We agree, and we changed the text in the next sections:
- Figure 4. The red arrow in G indicates vimentin (green) localized to the connective tissue surrounding the acini, and SMA (yellow) localized to the smooth muscle layer encircling the β-catenin (red) -positive acinar epithelial cells. The red arrow in H indicates the colocalization of the epithelial marker β-catenin (red) and mesenchymal markers SMA (yellow) and vimentin (green) in DC sweat glands.
- Discussion lines 213-216; The occurrence of EMT in DC-associated sweat glands was further confirmed by the colocalization of the mesenchymal and epithelial markers marker vimentin and β-catenin in the sweat gland epithelial compartment. Interestingly, no increase in vimentin or SMA was detected by our proteomic analysis.
9. On lines 173 and 174, it is stated that there was an upregulation of TGFBI in the DC-associated sweat gland. It was not clear whether this is the new data (proteomics data they have generated) or is it something previously published work they have stated there.
Response: We have clarified this point, and the same statement is shown next:
- In line with the anticipated activation of TGF-β signaling, our proteomic analysis detected an upregulation of TGFBI in DC-associate sweat glands.
10. In the 3rd paragraph of the discussion (lines 175-180), where they have connected the β-Catenin pathway with PI3K/AKT, RhoA and E-cadherin, it will be good if they discuss these scenarios in the context of their proteomics pathway analysis data.
Response: We have added some additional data that reinforces our hypothesis in the manuscript. The next sentence was in the text:
- We observed the downregulation of Ras suppressor-1 (RSU1), and the upregulation of Rab23, Rab8A, and RhoC (members of Ras superfamily) as well as several integrins, including ITGAV in our proteomic results, which indicate the activation of non-canonical-TGF-β-signaling pathway most probably through Ras oncogene [28-30].
11. On lines 200 and 201, two “that’ need to be added after “TGFBI” and after “proteins”, an ‘N” should be added to “POST”.
Response: We agree, and we changed this in the text.
12. An illustration of the signaling pathways uncovered in this study with all the feedback loops should be included.
Response: We agree, and we added a new figure (Figure 5) that summarizes the findings of the study.
Reviewer 2 Report
In this paper the authors investigated, by the proteomic analysis and immunofluorescence microscopy, the pathogenesis of the Dupuytren’s contracture (DC), which is a chronic and progressive fibro proliferative disease localized to the palmar fascia and leading to the loss of finger functions. They observed that the sweat gland epithelium undergo epithelial-mesenchymal transition associated to a downregulation of E-cadherin, increase of VIM and upregulation of various collagen types. They suggest that sweat gland cells, partially acquiring a mesenchymal phenotype, could release signals that stimulate and initiate tissue fibrosis inside the sweat glands themselves, transforming the epithelial cells into a source of mesenchymal cells or new myofibroblasts. This paper is well written, show nice pictures and may be actual because underlines the interplay between epithelial cells (sweat glands cells) and extracellular matrix, with particular attention to an EMT leading to fibrosis. Authors reported many limitations of this paper at the end of the Discussion section; however, even though the conclusions are supported by many data, this paper does not really demonstrate a direct transformation of sweat gland epithelial cells into myofibroblasts producing collagen which support fibrosis; authors have to discuss this limit also suggesting which techniques, in future investigations, could demonstrate what they hypothesize.
Minor suggestions:
Line 53, page 2: “We used immunofluorescence microscopy to corroborate our proteomic analysis, and we concluded that epithelial-mesenchymal tran sition (EMT) occurring in sweat glands connected to increased synthesis of connective tissue growth factor (CTGF) potentially participates in the progression of DC.”
Do not explicitly reveal in advance the conclusion of your investigation. Please change your sentence in a similar to this:
We used immunofluorescence microscopy to corroborate our proteomic analysis, and demonstrate that epithelial-mesenchymal -transition (EMT) occurring in sweat glands connected to increased synthesis of connective tissue growth factor (CTGF) could potentially participates in the progression of DC.
Line 58, page 2: “First, to collect material for proteomics analysis, we used the protocol previously established by Gao, et al. (2014) to isolate sweat glands from human skin.”
It could be better: To collect material for proteomics analysis by isolating sweat glands from human skin, we followed the protocol previously established by Gao, et al. (2014).
Line 60, page 2: The proteome analysis identified 4041 proteins, among which 641 were immunoglobulins and 128 cytokeratins, which were eliminated from further analysis.
Line 60, page 2: “extracellular matrix organization”..
Change in: “ECM organization”… Please, use this abbreviation in the other sentences of the paper.
Line 70, page 2: “a connected network of consisting of”
why not: a connected network consisting of……
Line 93, page 2: “corroborating well the results”
well corroborating the results
Line 112, page 5: the adherent junction pathway
the adherent junction pathway,
Line 162, page 7: that CFTGF was
Change in: that CFTGF is
Author Response
We wish to thank the reviewers for very constructive suggestions that helped to considerably increase the quality of the manuscript.
Please find the point-to-point answers to the reviewers’ criticisms below.
Reviewer #2:
- Line 53, page 2: “We used immunofluorescence microscopy to corroborate our proteomic analysis, and we concluded that epithelial-mesenchymal tran sition (EMT) occurring in sweat glands connected to increased synthesis of connective tissue growth factor (CTGF) potentially participates in the progression of DC.”
Do not explicitly reveal in advance the conclusion of your investigation. Please change your sentence in a similar to this:
Response: We used immunofluorescence microscopy to corroborate our proteomic analysis and demonstrate that epithelial-mesenchymal -transition (EMT) occurring in sweat glands connected to increased synthesis of connective tissue growth factor (CTGF) could potentially participate in the progression of DC. We agree to the comment and change this statement in the text.
- Line 58, page 2: “First, to collect material for proteomics analysis, we used the protocol previously established by Gao, et al. (2014) to isolate sweat glands from human skin.”
It could be better: To collect material for proteomics analysis by isolating sweat glands from human skin, we followed the protocol previously established by Gao, et al. (2014).
Response: We agree and we changed this in the text.
- Line 60, page 2: The proteome analysis identified 4041 proteins, among which 641 were immunoglobulins and 128 cytokeratins, which were eliminated from further analysis.
Response: We agree and changes were made in the text.
- Line 60, page 2: “extracellular matrix organization”.
Change in: “ECM organization”… Please, use this abbreviation in the other sentences of the paper.
Response: We agree and we used this abbreviation in the other sentences of the paper.
- Line 70, page 2: “a connected network of consisting of”
why not: a connected network consisting of……
Response: We agree and we changed this in the text.
- Line 93, page 2: “corroborating well the results”
well corroborating the results
Response: We agree and we changed this in the text.
- Line 112, page 5: the adherent junction pathway
the adherent junction pathway,
Response: We agree and we changed this in the text.
- Line 162, page 7: that CFTGF was
Change in: that CFTGF is
Response: We agree and we changed this in the text.
Round 2
Reviewer 1 Report
The authors have addressed the comments of this reviewer satisfactorily. revised manuscript is improved. The original figure 3 should be removed. When possible, new colors reflecting merging of two or more IHC-related stains should be indicated in the figure legends where appropriate.
Author Response
We wish to thank the reviewers for very constructive suggestions that helped to considerably increase the quality of the manuscript.
Please find the point-to-point answers to the reviewers’ criticisms below.
Reviewer #1:
The authors have addressed the comments of this reviewer satisfactorily. revised manuscript is improved. The original figure 3 should be removed. When possible, new colors reflecting merging of two or more IHC-related stains should be indicated in the figure legends where appropriate.
Response: We agree, the original figure 3 has been removed, and we have added information in the figure legends 3 and 4 to reflect the merging colors.
Figure 3. Sweat glands of Dupuytren’s contracture samples present low levels of E-cadherin. Immunofluorescence analysis of control sweat gland acini (A, E, I, M), and duct (B, F, J, N) DC sweat gland acini (C, G, K, O) and duct (D, H, L, P) samples were stained for CTGF (A—D), β-catenin (E—H), E-cadherin (I—L) and merged (M—P). E-cadherin and β-catenin colocalization (orange color). Scale bar: 50 µm. Quantitative analysis of E-cadherin (Q), β-catenin (R) levels in control and DC samples, and E-cadherin/β-catenin ratio (S). The duct in N is marked with a white arrow. Results are presented as mean ± SD and compared to control samples. * p < 0.05.
Figure 4. Dupuytren’s contracture samples with extensive stage of fibrosis present cells with in-termediate epithelial-mesenchymal transition phenotype. Immunofluorescence analysis of control (A, C, E, G) and DC (B, D, F, H) samples were stained for β-catenin (A, B), vimentin (C, D), and smooth muscle actin (E, F), and merged (G, H). Red arrow in G indicates vimentin (green) localized to the connective tissue surrounding the acini, and SMA (yellow) localized to the smooth muscle layer encircling the β-catenin (red) -positive acinar epithelial cells. Red arrow in H indicates the colocalization (dark yellow) of the epithelial marker β-catenin (red) and mesenchymal markers SMA (yellow) and vimentin (green) in DC sweat glands. Scale bar: 50 µm.
Reviewer 2 Report
Line 54:: To avoid repetition change:
“eventually form the DC nodule [7].” in “eventually forming the DC nodule [7]”.
Line 276: To avoid repetition change:
“and apical dark cells in acini, and epithelial cells in duct” in “and apical dark cells in acini, as well as epithelial cells in duct.”
Author Response
We wish to thank the reviewers for very constructive suggestions that helped to considerably increase the quality of the manuscript.
Please find the point-to-point answers to the reviewers’ criticisms below.
Reviewer #2:
Line 54:: To avoid repetition change:
“eventually form the DC nodule [7].” in “eventually forming the DC nodule [7]”.
Line 276: To avoid repetition change:
“and apical dark cells in acini, and epithelial cells in duct” in “and apical dark cells in acini, as well as epithelial cells in duct.”
Response: We agree and we changed this in the text.